# Complexes of Sodium Pectate with Nickel for Hydrogen Oxidation and Oxygen Reduction in Proton-Exchange Membrane Fuel Cells

**DOI:** 10.3390/ijms232214247

**Published:** 2022-11-17

**Authors:** Irek R. Nizameev, Danis M. Kadirov, Guliya R. Nizameeva, Aigul’ F. Sabirova, Kirill V. Kholin, Mikhail V. Morozov, Lyubov’ G. Mironova, Rustem R. Zairov, Salima T. Minzanova, Oleg G. Sinyashin, Marsil K. Kadirov

**Affiliations:** 1Arbuzov Institute of Organic and Physical Chemistry, FRC Kazan Scientific Center, Russian Academy of Sciences, 420088 Kazan, Russia; 2Department of Nanotechnologies in Electronics, Kazan National Research Technical University Named after A.N. Tupolev-KAI, 420111 Kazan, Russia; 3Department of Physics, Kazan National Research Technological University, 420015 Kazan, Russia; 4A.M. Butlerov Chemistry Institute, Kazan Federal University, 420008 Kazan, Russia

**Keywords:** proton exchange membrane fuel cell, membrane electrode assembly, oxygen reduction reaction, hydrogen oxidation reaction, coordination biopolymers

## Abstract

A number of nickel complexes of sodium pectate with varied Ni^2+^ content have been synthesized and characterized. The presence of the proton conductivity, the possibility of the formation of a dense spatial network of transition metals in these coordination biopolymers, and the immobilization of transition ions in the catalytic sites of this class of compounds make them promising for proton-exchange membrane fuel cells. It has been established that the catalytic system composed of a coordination biopolymer with 20% substitution of sodium ions for divalent nickel ions, Ni (20%)-NaPG, is the leading catalyst in the series of 5, 15, 20, 25, 35% substituted pectates. Among the possible reasons for the improvement in performance the larger specific surface area of this sample compared to the other studied materials and the narrowest distribution of the vertical size of metal arrays were registered. The highest activity during CV and proximity to four-electron transfer during the catalytic cycle have also been observed for this compound.

## 1. Introduction

Fuel cells involving the use of hydrogen as an energy carrier efficiently generate electricity, water, and heat and constitute an energy-efficient and environmentally friendly power source [1]. Among them, proton-exchange membrane fuel cells (PEMFCs) have been developed especially for use in vehicle engines and portable electronics because of their high-energy density, low operating temperature, quiet operation and quick start-up [2,3]. Highly dispersed platinum on a carbon carrier was used as anode and cathode catalysts for the oxygen reduction reaction (ORR) and the hydrogen oxidation reaction (HOR), respectively. The HOR rate at Pt/C of the anode is rather high because there is no need to dissolve hydrogen in water, it is decomposed into proton and electron directly on the solid electrode surface; this process is limited only by the proton diffusion rate. Correspondingly, the amount of Pt, which is expensive to extract and sparsely occurring in the Earth’s crust, can be reduced to 0.05 mg per 1 cm^2^ [4]. Additionally, the ORR rate in the membrane-electrode assembly (MEA) of PEMFC is generally bounded by the limited solubility of gaseous oxygen in water [5], and almost an order of magnitude larger amount of Pt (0.4 mg cm^−2^) is necessary to achieve an acceptable fuel cell capacities. Furthermore, the susceptibility of Pt to dissolution, coalescence, and subsequent flushing out from the active zone of the catalysis adversely affects the stability and durability of PEMFCs on Pt/C catalysts [6].

This may explain why so much attention has been paid to the development of molecular non-platinum ORR catalysts, and considerable progress has been made here [7,8,9,10]. To ensure that fuel cells with non-platinum ORR catalysts are as competitive as their platinum-containing analogues, in the most successful PEMFCs from the perspective of achieving high-power densities, various methods have been used to expand the number of active centres per unit volume through pyrolysis, or compounds with more catalytic centres have been taken initially. These are tetraphenylporphyrins of cobalt and iron on amorphous silica [7], iron cations coordinated with pyridine nitrogens in the voids of graphite sheets [8], a polypyrrole composite catalyst with two-coordinated cobalt [9], as well as polyaniline CoFe-C composite ORR catalysts [10], obtained by polymerization of aniline in a mixture of carbon, cobalt and iron ions and subsequent heat treatment.

The HOR on the platinum-carbon catalysts on the anode side proceeds efficiently and exhibits no serious kinetic limitations like the ORR on the cathode side. There are still the problems of precious metal scarcity, their high price, and the degradation of platinum-carbon catalysts. Reducing the amount of Platinum on the anode has an adverse impact on the long-term stability of the fuel cell, decreasing the effective operating time of the fuel cell on this catalyst.

As a promising substitute for conventional catalysts based on platinum and other precious metals, various transition metal (TM) ions (Fe, Co, Mo, Ni, V, Cu, etc.) have been used to develop advanced electroactive materials. Among these TMs, nickel has emerged as one of the most promising components due to its excellent electronic properties and the expected synergistic effect of drastically altering the material surface properties to foster electrocatalysis [11].

Recently, as a cost-effective and efficient substitute for precious metals, nickel-based electrocatalysts in the form of nickel foams [12], alloys [13], nitrides [14], phosphides [15], oxides [16], metal-organic frameworks (MOF) [17], metal complexes [18,19], coordination biopolymers [20,21], etc. have demonstrated electrocatalytic activity and overall stability towards the ORR, HOR, OER and HER. The high synergistic effect arising between Ni and neighboring atoms results in the significantly better adsorption properties of the surface, potentially leading to improved electrocatalytic properties of the obtained materials. Due to their low price, high content, excellent strength, better ductility, high corrosion resistance, good thermal conductivity and high electrical conductivity, nickel-based materials have been extensively researched for their electrochemical applications.

While the majority of well-known nickel compounds are divalent, it is capable of assuming other valences (from −1 to +4), which makes it very sensitive to the various effects of electronic transitions. Due to these extremely interesting electronic properties, high conductivity and thermal stability, Ni is widely used for the development of electrocatalytic materials.

This article serves as a continuation of our series of works on the application of coordination biopolymers, namely, complexes of natural pectin polysaccharides with nickel, as the ORR and the HOR electrocatalysts for the PEMFCs. Complexes of natural pectin polysaccharides have attracted our attention because they constitute stable biopolymers. Pectin is primarily derived from citrus and apple peels, and it also has a number of industrial applications associated with its gelling properties [22]. Pectin-based, ethanolamine [23] and diethanolamine [24] modified polymer electrolyte membranes for the fuel cells have also been reported. This indicates that pectin materials exhibit proton conductivity, an important property of PEMFC electrocatalytic systems. Secondly, the main carbohydrate chain of pectin polysaccharides is composed of 1,4-linked α-D-galacturonic acid residues, the complexes with divalent transition metal ions of which form a spatial network with a dense arrangement of metal ions [25,26]. The third attractive feature of the developed catalysts on pectin polysaccharide complexes stems from the fact that these compounds are poorly soluble in water, merely swelling in it and being immobilized [20,21,22], thus, ensuring the possible stability of the diagnostic characteristics of these catalysts in the composition of MEAs of the PEMFCs.

## 2. Results and Discussion

### 2.1. Synthesis of the Sodium Pectate Complex with Nickel

The synthesis of the complexes was performed according to the method described elsewhere [27]. The synthesis scheme is shown in Figure 1. To ensure the solubility of the desired products, initially a sodium pectate (NaPG) with a degree of salt formation of 100% was obtained by treating citrus pectin with alkali at controlled pH values with a titrimetric transition from the slightly acidic (pH 3.8) to the slightly alkaline region (pH 8.5–9.0). A 2-L flask was filled with 1.5 L of distilled water, 40 g of citrus pectin of Classic C-401 brand, produced by Herbstreith & Fox (Neuenbürg, Germany) (the degree of esterification is 65%) was poured into it and heated with constant stirring on a magnetic stirrer to 50–60 °C until the complete dissolution of pectin. In parallel, 5.0 g of alkali (NaOH) is dissolved in 100 mL of distilled water and added in small portions to the pectin solution at controlled pH values. Afterwards, the volume of the reaction mixture is brought up to two litres and NaPG is then synthesized for 2 h at a temperature of 50–60 °C and under constant stirring.

The state of carboxyl groups was controlled by IR spectroscopy (Appendix A, Table 1) in the range of stretching vibrations of the COO– group (1600–1800 cm^−1^). The completion of the reaction was indicated by the disappearance of the absorption band of stretching vibrations ν(C=O) of the carboxyl group at 1745–1750 cm^−1^ and the appearance of the absorption band of stretching vibrations ν (C=O) at 1600–1650 cm^−1^, characteristic of the ionic form.

The resulting sodium pectate was the initial ligand for the synthesis of metal complexes, then the desired compounds were synthesized by the reaction of ligand exchange of Na^+^ ions for Ni^2+^ cations. The experiments were planned in such a way as to create a rarefied three-dimensional structure of complexes with a relatively low degree of substitution of sodium ions for d-metals while retaining most of the sodium ions in the salt form as a part of the composition of the polymer complex to ensure its water-soluble properties (within 5–25% of divalent nickel metal relative to the initial content of monovalent sodium).Nickel salt solutions in 400 mL distilled water with masses of 0.0–1.148 g as indicated in the third column of Table 2 were added to 400 mL of sodium pectate solution and corresponding coordination polymer have been synthesized at 50–60 °C for 15–20 min. The reaction mixture is cooled off to room temperature and the complex is precipitated with a double volume of ethanol. The precipitate is separated by centrifugation and freeze-dried on an Alpha 1–2 LD dryer. In terms of organoleptic properties, the resulting Ni(n%)-NaPG compounds are amorphous greenish powders, soluble in water at 50–60 °C up to 2% concentration. The patterns of complexation in the sodium pectate–NiCl_2_·6H_2_O system were studied in the NiCl_2_·6H_2_O salt concentration range of 0.21–1.44 g/L (3rd column of Table 2).

### 2.2. Optical Atomic Emission Spectroscopy (AES) with Inductively Coupled Plasma

The concentrations of Na and Ni were determined (Table 3) in the complex extract using an optical emission spectrometer with synchronous inductively coupled plasma (ICP-OES) using the spectral lines of Na and Ni, 588.995 and 231.604 nm, respectively. The study of the elemental composition of the complex revealed that the expected metal content corresponds to the real one in the range of 8% error, which is quite an acceptable result.

### 2.3. Morphology Studies by TEM and AFM Methods

For a number of Ni(n%)-NaPG samples with varying degrees of nickel substitution (from 5% to 35%), the film structures formed during their deposition from the liquid phase onto the solid surface have been studied. Figure 2 shows TEM images (left column), AFM images (central column), and particle size distribution (right column) from the AFM images for the Ni(n%)-NaPG compounds indicated in the left column. TEM images demonstrate the formation of an organic film with the introduction of metal particles with a small size spread (a few nanometers). A high uniformity of the distribution of the array of metal particles in the volume of the organic matrix is observed. Upon that, it is possible to indicate the presence of self-organization and structuring of the system due to intermolecular interaction. It can be assumed that a certain cellular structure is formed, leading to a uniform redistribution of metal centers. A simplified model of this structure has been reported before [20].

For the Ni (5%)-NaPG and Ni (15%)-NaPG samples, there are areas of the film without metal filling. Perhaps because of these breaks in the organic film with rough edges, the vertical dimensions in the AFM images have large values (around 2 nm) and a wider vertical distribution. In contrast, the Ni (35%)-NaPG sample is characterized by the formation of thickenings of metal centres and their superposition in a number of regions. The most optimal substitution values in terms of the uniformity of the distribution of metal centers appear to be 20 and 25% [samples of Ni (20%)-NaPG and Ni(25%)-NaPG].

In many respects, AFM images replicate TEM results. On AFM images, metal centers can be identified from the topography of the films (light areas) and from the phase contrast image. Judging by the sharpness of the particle distribution histogram over vertical sizes, the most uniform distribution of particles is observed for Ni (20%)-NaPG. For further studies, Ni (15%)-NaPG, Ni (20%)-NaPG and Ni (25%)-NaPG coordination biopolymers were used. Ni (20%)-NaPG was selected as the sample with the most uniform vertical particle distribution, and Ni (15%)-NaPG and Ni (25%)-NaPG as the closest to the former in terms of nickel content.

### 2.4. CV in Acidic Solution

Figure 3 shows CVs with indication of the amplitudes of the cathodic peaks for the ORR on GC electrodes modified with coordination biopolymers Ni(15%)-NaPG (left), Ni(20%)-NaPG (center) and Ni(25%)-NaPG (right) in argon-saturated (dashed curves) or oxygen-saturated 0.5 M H_2_SO_4_ (solid curves). According to the CV curves of the studied coordination biopolymers, all of them exhibit activity of varying degrees regarding oxygen reduction. At the same time, the Ni(20%)-NaPG compound is approximately 1.5 times more active than the other studied samples. In solutions saturated with argon, there are no reduction peaks.

According to the CV curves of the studied coordination biopolymers, all of them exhibit activity of varying degrees regarding oxygen reduction. At the same time, the Ni(20%)-NaPG compound is approximately 1.5 times more active than the other studied samples. In solutions saturated with argon, there are no reduction peaks.

### 2.5. Electrochemical Stability during the ORR in Acidic Solution

Figure 4 shows the results of the potentiostatic stability tests of the Ni (20%)-NaPG/C catalyst on a CC electrode for 6000 s. The current in an argon atmosphere is stabilized at a minimum value of 0.0145 mA, while its value in an oxygen atmosphere is about 7 times greater and reaches a plateau in 3000 s. This means that oxygen is consistently reduced on the electrode surface modified by the catalyst under study. Long-term stability tests in PEMFC will be presented in future publications.

### 2.6. Study of the ORR Kinetics on a Rotating Disk Electrode Using the Koutecky–Levich Method

To study the kinetics of the oxygen electroreduction reaction on a rotating disk electrode, the Koutetcky–Levich equation was used [29,30]:1/I = 1/i_K_ +1/i_D_ = 1/i_K_ +1/Bnw^0.5^,(1)
where B = 0.62FD^2/^ υ^−1/6^c, i—current on the disk electrode; i_K_—kinetic current; i_D_—diffusion current; ω—disc electrode rotation speed (rad/s); n—the number of electrons involved in the electrochemical reaction; F—Faraday constant, C/mol; D—diffusion coefficient, cm^2^/s; υ—kinematic viscosity of electrolyte, cm^2^/s; c—oxygen concentration in solution, mol/cm^3^. The parameters given in Table 4 are suitable for our system. The reverse current for the Koutecky–Levich plot has been in all cases measured at the indicated potentials vs. Ag/AgCl.

To elucidate the ORR mechanism, voltammograms of O_2_ reduction using a rotating glassy carbon disk electrode modified with Ni(n%)-NaPG/C catalysts in 0.5 M aqueous oxygenated H_2_SO_4_ solution were obtained (Figure 5A,B and Appendix A). The Koutecky–Levich plots (Figure 5C,D and Appendix A) of the reverse current through the rotating disk electrode as a function of the square root of the inverse disk rotation speed indicated that per one catalytic cycle for the Ni (25%)-NaPG/C catalyst the number of electrons transferred per catalytic cycle is 2.2 [20], for Ni (15%)-NaPG/C it equals to 2.76, and for the Ni(20%)-NaPG/C composite − 3.11.

### 2.7. PEMFC Tests

Polarization studies have been performed by fabricating membrane electrode assemblies using Ni(n%)-NaPG/C as the cathode or anode electrocatalyst and commercial Pt/C as the anode or cathode electrocatalyst, respectively, at a load of 1 mg/cm^2^. Table 5 shows the values of open circuit voltage (OCV), maximum current and power densities obtained by various PEMFCs with Ni(n%)-NaPG and Pt based on Vulcan XC-72 carbon black. Figure 6 and Appendix A show H_2_/O_2_ PEMFC representations and the corresponding polarization curves, current and power density curves for MEAs Ni (15%)-NaPG/Nf/Pt, Pt/Nf/Ni(15%)-NaPG, Ni(25%)-NaPG/Nf/Pt, Pt/Nf/Ni(25%)-NaPG, Ni(20%)-NaPG/Nf/Pt and Pt/Nf/Ni(20%)-NaPG.

It is clear from polarization measurements that Ni (20%)-NaPG shows good HOR and ORR activity. Various aspects of the electrochemical oxidation of the Ni (25%)-NaPG compound studied by CV and EPR methods in the composition of a carbon paste electrode have been discussed in detail previously [21], and the considerable potential of the catalytic system based on it has been noted. Good activity and stability of this catalytic system have been observed. The performance of PEMFC Ni (20%)-NaPG/Nf/Pt in the hydrogen oxidation reaction indicates already a 4–5-fold increase in maximum specific current and maximum specific power compared to other studied catalytic systems with different nickel contents, including Ni (25%)-NaPG. One of the main reasons for the good catalytic activity of the Ni (20%)–NaPG/C biocomposite can be attributed to the value of the specific surface area (Table 6), which is larger compared to that of the other samples studied.

Another reason may be related to the narrow distribution of the vertical size of metal particle arrays in the volume of the organic matrix, which has been outlined above in the TEM-AFM section. Moreover, this size of 1 nm may itself play an integral role in the catalytic activity of the coordination biopolymers under study.

As can be seen from the CV curves of the studied coordination biopolymers, all of them exhibit activity of varying degrees with respect to the reduction of oxygen in an acidic liquid medium. At the same time, the Ni (20%)-NaPG compound turned out to be approximately 1.5 times more active than the other samples under study.

The catalytic activity in the ORR, in turn, is also determined by the number of transferred electrons per one catalytic cycle. For the Ni (20%)-NaPG/C composite, of all those studied, it turned out to be the closest to the optimal four-electron transfer for the ORR −3.11. For the Ni (25%)-NaPG/C catalyst, the number of electrons transferred per one catalytic cycle was 2.2 [20], for Ni (15%)-NaPG/C it was 2.76. This means that for the Ni (25%)- NaPG/C the ORR is only 10% four-electron, for Ni (15%)-NaPG/C it is 36%, and for Ni (20%)-NaPG/C it is already 55.5%.

Naturally, all these reasons causing different catalytic activity of the studied systems of coordination biopolymers can be combined with each other by cause-and-effect relationships and provide a complex of synergistic effects. These aspects of the findings may constitute the subject of special, more theoretical research in the future.

## 3. Materials and Methods

### 3.1. Synthesis of the Sodium Pectate Complex with Copper

We used citrus pectin of “Classic C-401” brand produced by Herbstreith & Fox (Germany) as an organic matrix for the introduction of nickel ions. The measured molecular weight of the citrus pectin is 17,6 kDa. NiSO_4_·5H_2_O, NaOH and other reagents with a purity of more than 99.9% have been used for the synthesis.

### 3.2. Fourier-Transform Infrared Spectroscopy (FTIR)

IR spectra were recorded on IR-Fourier spectrophotometer IRS-113 (Bruker, Billerica, MA, USA) with 1 cm^−1^ resolution in the range 400–4000 cm^−1^, the substance being pressed with KBr in tablets.

### 3.3. Inductively Coupled Plasma Optical Emission Spectroscopy

Na and Ni concentrations were identified in the complex extract using simultaneous inductively coupled plasma optical emission spectrometer (ICP-OES) model iCAP 6300 DUO by Varian Thermo Scientific Company equipped with a CID detector. Together, the radial and axial view configurations enable optimal peak height measurements with suppressed spectral noises. 10 mg of the complex powder was placed in 20 mL of a 0.2 molar solution of HNO_3_ to prepare extracts of the complexes. The concentrations of Na and Ni ions were determined by the spectral lines 588.995 and 231.604 nm, respectively. Sc has been used as internal standard (10 ppm in the sample), and all the standards have been produced by the Perkin Elmer corporation.

### 3.4. Specific Surface Area

The specific surface area of the coordination biopolymers has been measured on a QuantachromeNova 1200e device for the adsorption of nitrogen gas using the Brunauer, Emmett and Teller method (BET—method).

### 3.5. Electrochemistry

The electrochemical measurements of the Ni(n%)-NaPG coordination biopolymers were performed in a glass three-electrode electrochemical cell using an Elins P-20x potentiostat/galvanostat (Appendix A). The working electrode is a glassy carbon disc having a diameter of 3 mm, the counter electrode is a platinum plate and the reference electrode is a saturated silver chloride electrode. The studies were carried out in an aqueous solution of 0.5 M H_2_SO_4_. Oxygen or argon was supplied to the electrolyte from cylinders through a reducer with a fine adjustment valve and a glass capillary. Catalytic ink with a volume of 20 µL was applied to the surface of a glassy carbon disc electrode and allowed to dry completely. Drying time is always different, depending on the substance, on average, from 20 to 60 min. The solutions were bubbled with oxygen or argon for 20 min.

### 3.6. TEM

The transmission electron microscopy images were obtained with Hitachi HT7700, Japan. The images were acquired at an accelerating voltage of 100 kV. Samples were ultrasonicated in water for 10 min and then dispersed on 300 mesh copper grid with continuous carbon-formvar support film.

### 3.7. AFM

Microscopic images were captured by scanning probe microscopy MultiMode V (Veeco instruments Inc., Plainview, NY, USA) using silicon cantilevers RTESP (Veeco instruments Inc., United States) with nominal spring constants of 40 N/m and tip curvature radius of 10–13 nm. Images were obtained using contact AFM techniques.

### 3.8. Fuel Cell Tests

The catalytic ink was prepared according to the following procedure: 4 mg of organometallic catalyst and 40 mg of Vulcan XC-72 were added to 1.5 mL of IPA and 1.5 mL of deionized water. The ink was first sonicated for 15 min; then 320 µL of 10 wt% Nafion^®^ solution (Aldrich, St. Louis, MO, USA) was added to the ink and sonicated again for 1 h. The E-TEK (Pt_20_/C) catalytic ink was prepared according to the almost similar procedure: 20 mg of E-TEK were added to 8.4 mL of IPA, then ink was sonicated for 15 min, and then 81 µL of 10 wt% Nafion^®^ solution was added to the ink and sonicated again. The inks were deposited on carbon paper gas diffusion layer (GDL) Sigracet^®^ 25CC. MEA was obtained by hot-pressing of GDLs on both sides of the Nafion^®^ 115 membrane at 90 °C with the load of about 300 lbs during 4 min. Polarization curves were obtained using the mechanical test station ElectroChem (United States) with the gas flow and pressure control system MTS-A-150 and the electronic load unit ECL-150. The surface concentration of nickel catalyst was 1 mg/cm^2^. The surface concentration of Pt was 1 mg/cm^2^ (5 mg Pt_20_/C on 1 cm^2^). The area of the MEA was 1 cm^2^. The MEA was tested in standard 1 cm^2^ PEMFC of ElectroChem Ink. (US, catalogue number—FC-01-02) (Appendix A). During the measuring of the polarization curves, the load was gradually increased until the maximum voltage (open-circuit voltage). The cell temperature was 80 °C. Flow rates of hydrogen and oxygen at 1.7 mL s^−1^ and 3.3 mL s^−1^, respectively. Anode and cathode gases humidified at 25 °C. The back pressure of the gases is 2.0 and 3.4 atm on the anode and cathode sides of the cell, respectively.

## 4. Conclusions

During PEMFC tests carried out on a series of composites of sodium pectate complexes with nickel as catalysts for hydrogen oxidation and oxygen reduction reactions in proton-exchange membrane fuel cells with 5, 15, 20, 25, 35% substitution of sodium ions by divalent nickel ions, the coordination biopolymer with 20% substitution demonstrated better catalytic performance, expressed in values of the maximum specific current and maximum power density. Among the studied possible reasons for the improvement in performance, the large specific surface area of this sample compared to the other studied materials is considered. Another reason may be the narrowest distribution of the vertical size of arrays of metal particles in the volume of the organic matrix of the sample with 20% substitution. In experiments on CV electrochemical reduction of oxygen in an aqueous solution of 0.5 M H_2_SO_4_, the Ni(20%)- NaPG compound is about 1.5 times more active than the other studied samples. Finally, the number of electrons transferred per catalytic cycle of the ORR for Ni(20%)—NaPG is closest to the four-electron process compared to the number per catalytic cycle for other systems in this series.

## Figures and Tables

**Figure 1 ijms-23-14247-f001:**
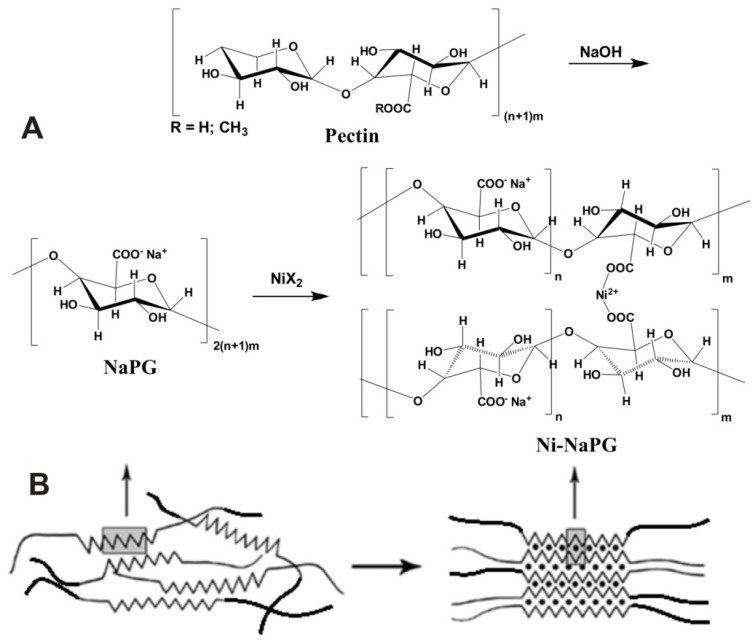
Schemes for the synthesis of pectin polysaccharide complexes (*n* = 3–10; *m* = 10–35) with nickel (**A**) and the formation of polymer-complex structures according to the “egg-box” [28] model (**B**—right).

**Figure 2 ijms-23-14247-f002:**
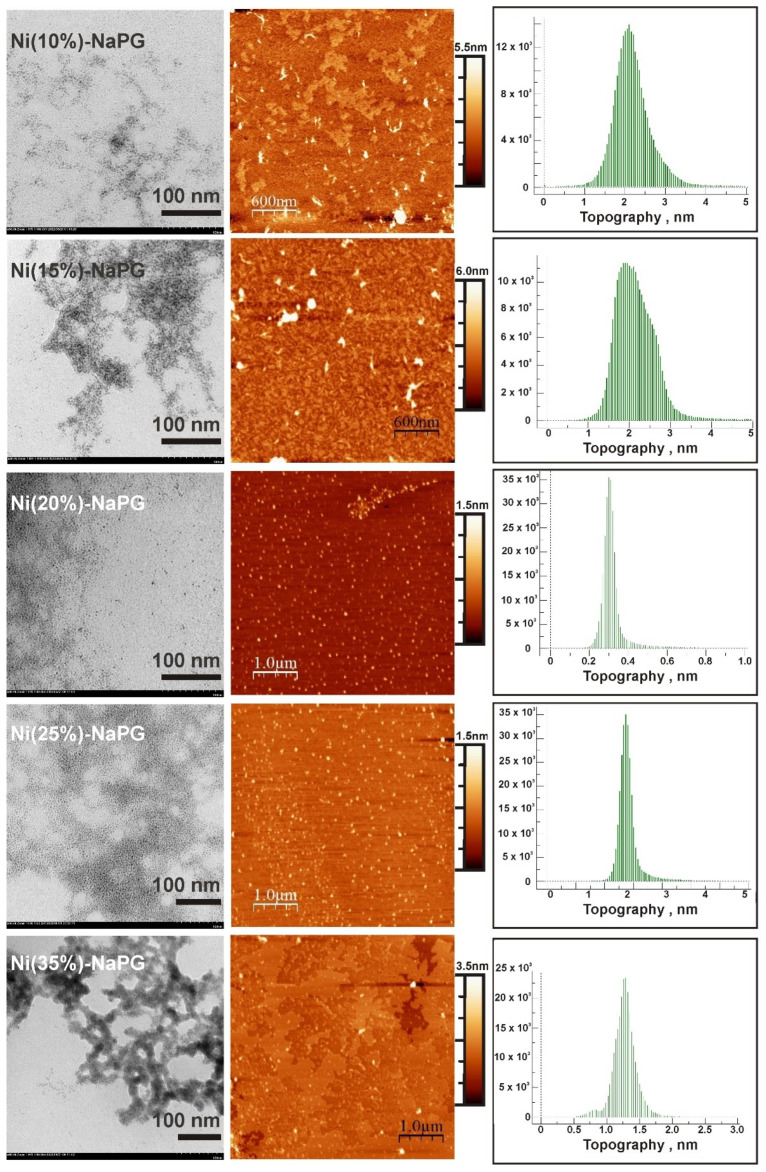
TEM images (**left column**), AFM images (**central column**) and particle size distribution (**right column**) from the AFM images for the Ni(n%)-NaPG compounds indicated in the left column.

**Figure 3 ijms-23-14247-f003:**
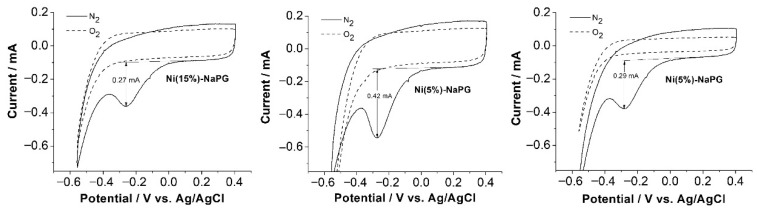
CV indicating amplitudes of cathodic peaks for the ORR on GC electrodes of Ni (15%)-NaPG (**left**), Ni(20%)-NaPG (**center**), and Ni(25%)-NaPG (**right**) coordination biopolymers in a nitrogen-protected environment (dashed curves) or oxygenated 0.5 M H_2_SO_4_ (solid curves) at a scan rate of 50 mV s^−1^; GC electrode area: 0.0707 cm^2^.

**Figure 4 ijms-23-14247-f004:**
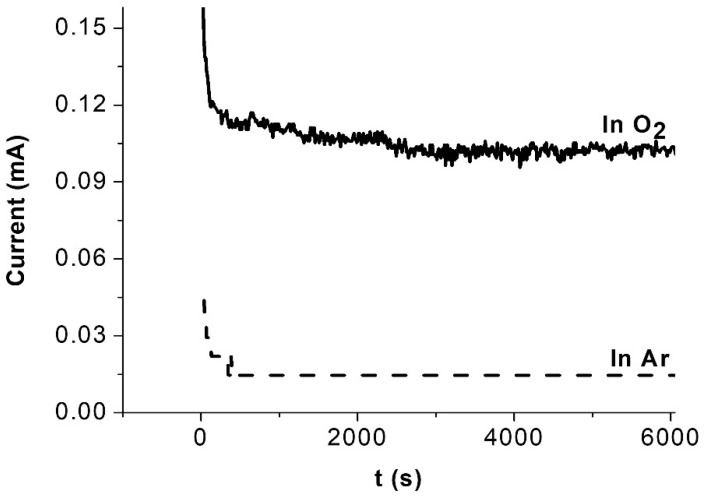
Potentiostatic stability tests of the Ni (20%)-NaPG/C catalyst on a CC electrode (S = 0.0707 cm^2^) in 0.5 M H_2_SO_4_ for 6000 s in both oxygen (solid, thick curve) and argon (dashed curve) saturated solutions at the same applied potential (E = −0.15 V vs. Ag/AgCl) with 0.05 mg of the catalyst.

**Figure 5 ijms-23-14247-f005:**
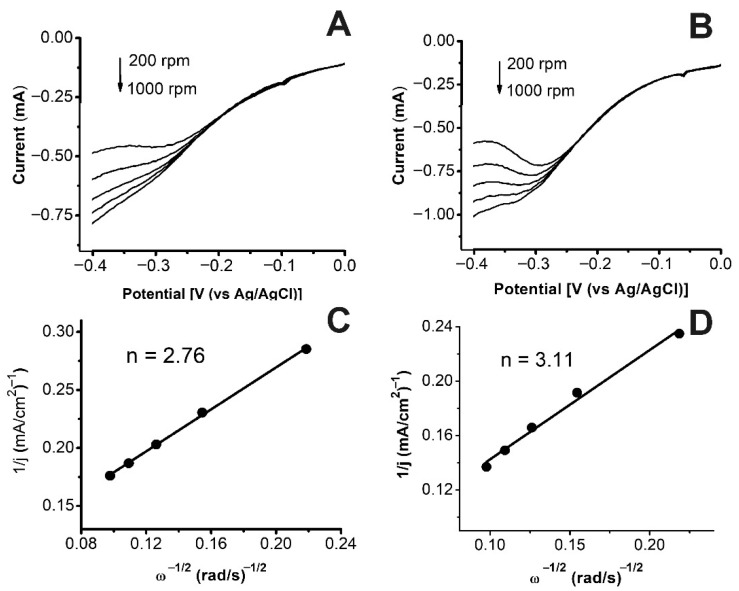
The first halves of the CVs (**A**,**B**) measured at different speeds of the RDE, the dependencies of Koutecky–Levich at − 0.4 V (**C**,**D**), corresponding to the Ni(15%)-NaPG/C (**A**,**C**) and Ni(20%)-NaPG/C (**B**,**D**) composites in 0.5 M H_2_SO_4_ solution.

**Figure 6 ijms-23-14247-f006:**
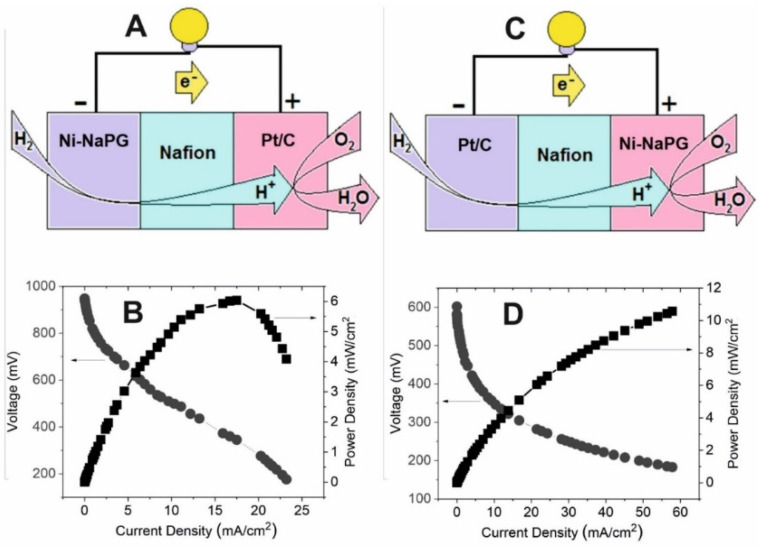
Representations (**A**,**C**) of the H_2_/O_2_ PEMFC and polarization and power curves at 80 °C for MEAs Ni(20%)-NaPG/Nf/Pt (**B**) and Pt/Nf/Ni(20%)-NaPG (**D**).

**Table 1 ijms-23-14247-t001:** The positions of the maxima of the IR spectra main characteristic bands (cm^−1^) of the pectin, sodium polygalacturonate and sodium polygalacturonate complexes with nickel.

Type of Oscillation	Pectin [Herbstreith & Fox KG (Germany)]	NaPG-Sodium Polygalacturonate	Sodium Polygalacturonate Complex with Nickel			
			Ni(5%)-NaPG	Ni(15%)-NaPG	Ni(20%)-NaPG	Ni(25%)-NaPG
ν (OH)	3430	3439	3443	3437	3443	3432
ν (CH)	2930	2944	2930	2930	2940	2929
ν (C=O)	1747	-	-	-	-	-
ν COO-	-	1614	1614	1621	1618	1620
δ (CH)	1331	1334	1332	1335	1332	1332
δ (OH)	1234	1242	1242	1240	1240	1241
ν (C-O-C)	1148	1147	1147	1148	1147	1149
ν (C-C) (C-O)	1105	1101	1102	1101	1101	1102
ν (C-C) (C-O)	1017	1013	1016	1018	1018	1019

**Table 2 ijms-23-14247-t002:** Samples of the synthesized coordination polymers, the degree of substitution of sodium ions for nickel ions, the mass of the initial salt per 400 mL of sodium pectate solution.

Sample	Degree of Substitution of Sodium Ions for Nickel Ions (%)	Mass of NiCl_2_·6H_2_O (g)	Concentration of NiCl_2_·6H_2_O (g/L)
NaPG	0.0	0.0	0
Ni(5%)-NaPG	5.0	0.164	0.21
Ni(15%)-NaPG	15.0	0.492	0.62
Ni(20%)-NaPG	20.0	0.656	0.82
Ni(25%)-NaPG	25.0	0.820	1.03
Ni(35%)-NaPG	35.0	1.148	1.44

**Table 3 ijms-23-14247-t003:** The content of Na and Ni in the synthesized Ni(n%)-NaPG coordination polymers determined using an AES iCAP 6300 DUO (Varian, Palo Alto, CA, USA) with synchronous inductively coupled plasma.

Coordination Biopolymer	Elemental Content (%) (Na + Ni = 100%)		Expected Content of Ni (%)	Deviation of Actual Ni Content from Expected (%)
	Na (588.995 nm)	Ni (231.604 nm)		
Ni(5%)-NaPG	95.4	4.6	5.0	8.0
Ni(15%)-NaPG	85.0	15.0	15.0	0
Ni(20%)-NaPG	81.0	19.0	20.0	5.0
Ni(25%)-NaPG	75.9	24.1	25.0	3.60
Ni(35%)-NaPG	66.8	33.2	35.0	5.14

**Table 4 ijms-23-14247-t004:** Parameter values for the Koutecky–Levich equation.

Electrolyte	D, cm^2^/s	υ, cm^2^/s	C_O2_, mol/cm^3^	B, mAs^0.5^cm^−2^	F, C/mol
0.5 M H_2_SO_4_	1.8 × 10^−5^	0.01	1.13 × 10^−6^	0.4	96,484.6

**Table 5 ijms-23-14247-t005:** Open circuit voltage (OCV), maximum current and power densities generated by the various PEMFCs based on Ni(n%)-NaPG on carbon black (Vulcan XC-72) and carbon black-supported Pt.

Entry	Ni Content %	Anode Catalyst	Cathode Catalyst	OCV [mV]	Maximum Current Density [mA cm^−2^]	Maximum Power Density [mW cm^−2^]
1 ^[a]^	25	Pt/C	Ni-NaPG/C	710	59	5.9
2	20	Pt/C	Ni-NaPG/C	602	57.8	10.58
3	15	Pt/C	Ni-NaPG/C	730	15.33	2.76
4 ^[b]^	25	Ni-NaPG/C	Pt/C	960	5.2	1.5
5	20	Ni-NaPG/C	Pt/C	929	23.21	6.03
6	15	Ni-NaPG/C	Pt/C	830	4.89	1.54
7	100	Pt/C	Pt/C	1	1280	324

^[a]^ From [18]. ^[b]^ From [21].

**Table 6 ijms-23-14247-t006:** Specific surface area of the coordination biopolymers, measured on the QuantachromeNova 1200e by the adsorption of nitrogen gas using the Brunauer, Emmett and Teller method (BET—method).

Coordination Biopolymer	Sample Degassing			Specific Surface according to BET, m^2^/g
	Time, h	Temperature, °C	Vacuum, Pa	
Ni(15%)-NaPG	5	100	2	0.135
Ni(20%)-NaPG	5	100	2	0.147
Ni(25%)-NaPG	5	100	2	0.093

## Data Availability

Not applicable.

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
