# Peer review of "Complexes of Sodium Pectate with Nickel for Hydrogen Oxidation and Oxygen Reduction in Proton-Exchange Membrane Fuel Cells"

_ijms, 2022, doi:10.3390/ijms232214247_

Round 1

Reviewer 1 Report

In my opinion, this is an interesting paper for the community of the IJMS, relevant for the topic of fuel cells. In this paper, the utilization of a non-precious metal material is studied as alternative to Pt/C, which intrinsically is a hot topic in the field, given the price and availability of this the platinum (or similar materials). In my opinio, the paper is suitable for publication after some minor revisions:

1. Line 68, what do the authors mean with "exciting"? Could you replace by a smarter word? Perhaps, excellent, adequate...

2. Line 104, which is the well-known method? You may use "...according to the method described elsewhere [29-29,33-34].

3. Lines 134-137, please rewritte the sentence. It is incomprehensible.

4. How do you determine the parameters listed in Table 4?

5. A reference to long-term stability tests should be included in the manuscript given their importance for future applications. Perhaps as suggestions.

Author Response

In my opinion, this is an interesting paper for the community of the IJMS, relevant for the topic of fuel cells. In this paper, the utilization of a non-precious metal material is studied as alternative to Pt/C, which intrinsically is a hot topic in the field, given the price and availability of this the platinum (or similar materials). In my opinio, the paper is suitable for publication after some minor revisions:

  1. Line 68, what do the authors mean with "exciting"? Could you replace by a smarter word? Perhaps, excellent, adequate...

Response: “…nickel has emerged as one of the most promising components due to its excellent electronic properties…”

  1. Line 104, which is the well-known method? You may use "...according to the method described elsewhere [29-29,33-34].

Response: "...according to the method described elsewhere [29-29,33-34]…”

  1. Lines 134-137, please rewritte the sentence. It is incomprehensible.

Response: "nickel salt solutions in 400 ml distilled water with masses of 0.0-1.148 g as indicated in the third column of Table 2 were added to 400 ml of sodium pectate solution and corresponding coordination polymer have been synthesized at 50-60 oC for 15-20 minutes”

  1. How do you determine the parameters listed in Table 4?

Response: “Table 4. Parameter values for the Koutecki-Levich equation [32].”

  1. A reference to long-term stability tests should be included in the manuscript given their importance for future applications. Perhaps as suggestions.

      Response: “Figure 4 shows the results of the potentiostatic stability tests of the Ni (20%)-NaPG/C catalyst on a CC electrode for 6000 s. The current in an argon atmosphere is stabilized at a minimum value of 0.0145 mA, while its value in an oxygen atmosphere is about 7 times greater and reaches a plateau in 3000 s. This means that oxygen is consistently reduced on the electrode surface modified by the catalyst under study. Long-term stability tests in PEMFC will be presented in future publications.”

Reviewer 2 Report

The authors describe, the synthesis of Nickel Sodium Pectate for hydrogen oxidation and oxygen reduction in proton exchange membrane fuel cells.

In my opinion, this paper is not suitable for publication in this journal, the synthesis of the catalyst and applications have been described by the same authors (Inorg. Chem. Front., 2018,5, 780-784) reference 20.

Perhaps these studies are preferable to being published in another low impact factor journal.

On the other hand, I suggest analyzing the catalyst by XPS to see the nature of the nickel, is there nickel oxide?

Author Response

The authors describe, the synthesis of Nickel Sodium Pectate for hydrogen oxidation and oxygen reduction in proton exchange membrane fuel cells.

In my opinion, this paper is not suitable for publication in this journal, the synthesis of the catalyst and applications have been described by the same authors (Inorg. Chem. Front., 2018,5, 780-784) reference 20.

Perhaps these studies are preferable to being published in another low impact factor journal.

Response: The present study is aimed at solving an important problem - the development of efficient environmentally friendly proton-exchange fuel cells, namely the synthesis of Ni-NaPG nickel sodium polygolucturonates used as electrocatalysts in the hydrogen oxidation and oxygen reduction reactions.

We have been dealing with this problem in the last 5-7 years, so the positive results of exploratory studies that confirmed the promise of this direction were published earlier (Inorg. Chem. Front., 2018, 5, 780-784, reference 20).

Target product development is a complex task and requires a serious approach. It was found that a 25% degree of substitution of sodium ions for nickel ions is the threshold concentration for obtaining water-soluble pectin complexes. The synthesis of this coordination biopolymer was described in the article by Inorg. Chem. Front., 2018, 5, 780-784, Ref. 20.

Synthesis of subsequent coordination biopolymers, described in the published article, with 5%, 15%, 20%, 35% substitution of Na for Ni, of which Ni(5%)-NaPG, Ni(15%)-NaPG, Ni(20 %)-NaPG are water-soluble, and Ni(35%)-NaPG forms a gel, was of particular interest from the point of view of the water solubility effect of the catalysts on the studied catalytic properties. The study of the influence of this factor on the properties of the target products and their effectiveness was of interest. Therefore, a lot of work has been done to optimize the composition of the compounds, complexes with different contents of nickel ions have been obtained, and screening has been carried out to identify the most effective product.

On the other hand, I suggest analyzing the catalyst by XPS to see the nature of the nickel, is there nickel oxide?

Response: “XPS investigation of Ni(25%)-NaPG has been described in Supporting information of the reference 20. In the experimental diffraction pattern of the Ni(25%)-NaPG (fig. S4) there are no interference peaks corresponding to any crystalline form of nickel oxide whаt indicates its absence in the studied compounds.”

Round 2

Reviewer 2 Report

After the corrections requested by the 1st rapporteur and the arguments given by the authors, I accept the publication of the article.

Author Response

Dear reviewer, I send the edited version of the article according to your comments.